# Optimization of Thermal Conductivity vs. Bulk Density of Steam-Exploded Loose-Fill Annual Lignocellulosics

**DOI:** 10.3390/ma16103654

**Published:** 2023-05-11

**Authors:** Ramunas Tupciauskas, Andris Berzins, Gunars Pavlovics, Oskars Bikovens, Inese Filipova, Laura Andze, Martins Andzs

**Affiliations:** Latvian State Institute of Wood Chemistry, Dzerbenes 27, 1006 Riga, Latvia; andris.berzins@kki.lv (A.B.); pavlovichs@inbox.lv (G.P.); oskars.bikovens@kki.lv (O.B.); inese.filipova@kki.lv (I.F.); laura.andze@kki.lv (L.A.); martins.andzs@kki.lv (M.A.)

**Keywords:** wheat straw, reed, corn stalk, steam explosion, lignocellulosic biomass-based thermal insulation materials, thermal conductivity

## Abstract

Lignocellulosic biomass (LCB)-based thermal insulation materials available in the market are more expensive than conventional ones and consist mainly of wood or agricultural bast fibers which are primarily used in construction and textile industries. Therefore, it is crucial to develop LCB-based thermal insulation materials from cheap and available raw materials. The study investigates new thermal insulation materials from locally available residues of annual plants like wheat straw, reeds and corn stalks. The treatment of raw materials was performed by mechanical crushing and defibration by steam explosion process. Optimization of thermal conductivity of the obtained loose-fill thermal insulation materials was investigated at different bulk density levels (30–45–60–75–90 kg m^−3^). The obtained thermal conductivity varies in range of 0.0401–0.0538 W m^−1^ K^−1^ depending on raw material, treatment mode and a target density. The changes of thermal conductivity depending on density were described by the second order polynomial models. In most cases, the optimal thermal conductivity was revealed for the materials with the density of 60 kg m^−3^. The obtained results suggest the adjustment of density to achieve an optimal thermal conductivity of LCB-based thermal insulation materials. The study also approves the suitability of used annual plants for further investigation towards sustainable LCB-based thermal insulation materials.

## 1. Introduction

Thermal insulation materials are very important building construction materials with a general aim to reduce energy consumption of a building while simultaneously providing the indoor comfort temperature of 20 ± 1 °C. They are particularly important in the regions of the northern half of the globe with dominating low average yearly temperatures (<5 °C). In this case, the properly installed thermal insulation materials can significantly reduce greenhouse gas (GHG) emissions which are closely related to the reduction of carbon dioxide (CO_2_) [1].

Various thermal insulation materials are available on the market; however, up to 90% of them are made from non-renewable materials, such as plastic- or mineral-based fibers and foams [2]; these demand high energy consumption and cause significant detrimental effects on the environment, as well as recycling problems. Climate change policy set by the European Commission for reducing net GHG emissions [3] and the rising cost of energy resources force the search for new eco-effective thermal insulation materials. In the context of the Green Deal, the development of renewable lignocellulosic biomass (LCB)-based fiber materials from the residues of agriculture or recyclables (paper, cardboard or cotton-based textiles) is particularly acceptable, because they are intended to contribute to “green building” with nearly zero energy. Moreover, it was declared that the production of LCB-based materials has a chain of advantages such as low environmental impact, less energy consumption, low cost, low density, scalability, biodegradability and good insulation properties [4]. This and other studies also demonstrate a high potential of LCB-based thermal insulation materials obtained from a broad lignocellulosic feedstock to compete with conventional non-renewable thermal insulation materials by availability and related properties while simultaneously providing comfort, environmental friendliness of a building and energy efficiency [2,5,6,7,8].

The most abundant, locally available and low-cost agricultural residue is wheat straw (*Triticum aestivum*), which has been started to use in the form of bales with good insulation performance for the construction of green buildings throughout the world [7]. Some successive attempts were made to obtain insulation fiberboards from refined wheat straw pulp which were on the level of wood-based fiberboards [9]. Without cereal crops, corn (*Zea mays*) stalk could be the second most abundant residue amounting to 0.5 kg of biomass per dry corn grain produced, for which application is still limited [10]. The reed (*Phragmites australis*) wetland plant is also highly productive (up to 30 t per ha per year), locally available and cheap LCB in spite of its consideration for many applications including thermal insulation [2,11]. Still, there is a lack of information about the application of these raw materials for thermal insulation purposes by processing them into fibers.

It is well known that a steam explosion (SE) treatment using only water to generate a saturated steam is capable of converting raw LCB into a fibrous mass within a few minutes [12]. SE technology is recognized as one of the most appropriate and environmentally friendly pretreatments for the disruption of lignocellulosic structure to make value-added products like ethanol, methane, antioxidants, resins and cellulose nanofiber [13,14]. SE was advised to be used as an effective pretreatment of wood, improving gas permeability and the sound absorption capability to control the acoustical housing environment [15]. It has demonstrated good results obtaining homogeneous fibrous material from wheat straw by SE performed at 200 °C for 3 min [16] and from maize straw and miscanthus at 200 °C for 20 min [17]. Furthermore, the obtained bulk fiber mass has a good thermal conductivity and could be directly used for thermal insulation application without binders [18].

As it is known, very important factors influencing thermal conductivity are raw materials and the density of thermal insulation materials [8,19]. So, based on the above literature review and our previous studies [20,21], this paper continues the research of thermal insulation materials from locally available annual LCB such as wheat straw, water reed and corn stalk providing a deeper insight of the detected properties and optimizing the relationship between thermal conductivity and bulk density. The main purpose of the study was to find out an optimal bulk density of the investigated loose-fill LCB depending on SE conditions for further research providing other important properties like settling, water vapor diffusion, reaction to fire, mold fungi resistance and volatile emission.

## 2. Materials and Methods

### 2.1. Raw Lignocellulosics

Wheat straw—WS (grain-extracted from Limbaži district, Latvia), reeds (whole plant harvested in winter from Puzes Lake, Ventspils district, Latvia) and corn stalks—CS (fresh, ear/grain-extracted from the farm “Pauri”, Blome, Latvia) were used in the study as locally grown raw materials. The delivered raw materials were chopped in a knife mill (CM4000, LAARMANN, Roermond, The Netherlands) to pass a sieve with openings of Ø 30 mm.

### 2.2. SE Treatment

Based on the previous studies [20,21], the chopped raw LCB, first, was moisturized up to 80% of moisture content by immersing it in water. The moisture content (dry basis) of moisturized LCB was detected by a moisture analyzer (Precisa XM 120, Dietikon, Switzerland) following the standard (EN 14774-3) methodology [22]. The moisturized raw LCB was separately treated in a home-made SE device of original construction with a 0.5 L batch reactor at the constant temperature of 230 °C and varying residence time of 15 s, 30 s, 40 s and 50 s (Table 1). The used temperature during the treatment resulted in a pressure in the reactor of 30 ± 1 bar. The SE device includes a water boiler (6 L) to generate saturated steam, the reactor preheated by electricity for biomass treatment with saturated steam and a receiver (33 L) for biomass collection. The severity factor (R0) combining temperature (T, °C) and reaction time (t, min) was used to express the SE process from the following equation [23]:R0 = t × exp [(T − 100)/14.75].(1)

About 350 g (oven-dry mass) of raw lignocellulosic mass was treated by each SE mode. Immediately afterward, the treatment the biomass was collected and squeezed in a juice-like press to remove the liquid fraction, which contained 55 ± 8% of the treated wet biomass (SE-LCB).

### 2.3. Mechanical Foaming of SE-LCB

The liquid-squeezed SE-LCB samples were similar to a pressed wet pulp with a moisture content of 70 ± 5%. To convert the pressed LCB to a fluffy one, the wet biomass samples were mechanically processed by a self-made device through a system of two rotating cylinders (900 revolutions per minute) coupled with stainless steel wires as schematically shown in Figure 1a. Hereinafter, this process we call “foaming”. Initially, the wet biomass samples were foamed three times, then air-dried up to 90 min and foamed again for the last time at 1800 revolutions per minute.

The foamed samples were oven-dried at 60 °C to reduce the moisture content and then conditioned at a relative humidity of 60 ± 5% and a temperature of 20 ± 2 °C to achieve a constant mass before the testing. The fiber foaming process was done without any added external components, and the foamed biomass is shown in Figure 1b.

### 2.4. Characterization of Raw and SE-LCB

#### 2.4.1. Determination of Chemical Composition

Chemical components in the form of cellulose (glucan), hemicelluloses (sum of xylan, galactan, mannan, arabinan and acetyl groups), lignin and ash of raw and SE-LCB samples were detected following the analytical procedures described in Laboratory Analytical Procedures NREL/TP-510-42618 and NREL/TP-510-42622 [24,25]. For raw LCB samples, the lignin portion was summed as acid soluble and insoluble, while for SE-LCB samples only acid insoluble lignin was detected.

#### 2.4.2. Determination of Fourier Transform Infrared Spectra

Fourier transform infrared (FTIR) spectra of raw and SE-LCB samples were recorded in KBr (IR grade, Sigma Aldrich, Darmstadt, Germany) pellets by a Thermo Fisher Nicolet iS50 spectrometer (Waltham, MA, USA). The range of FTIR spectra was 4000–450 cm^−1^ with the resolution of 4 cm^−1^ and the number of scans 32. All spectra were normalized to the highest absorption maxima.

#### 2.4.3. Determination of Particle Size Distribution

To detect the particle size distribution of conditioned raw and foamed SE-LCB samples, 3 min fractionation was performed in a sieve column with the screen dimensions of 0.5, 1.0, 2.0, 3.0, 5.0 and 10 mm by Haver & Boecker (59032 Oelde, Germany) following the standard (EN 15149-1) methodology [26].

#### 2.4.4. Application of Light Microscopy

To evaluate the visible differences between the raw and SE-LCB samples, a light microscopy was used. The fiber surface and defibration level were observed by stereo microscope Stemi 508 (Carl Zeiss Microscopy GmbH, Jena, Germany) with double spot illumination at magnification of 0.63×. Before microscopic observation, each sample was placed in an open vial with the target bulk density of 60 kg m^−3^.

#### 2.4.5. Determination of Water Retention Value

Water retention value (WRV) is an empirical measure of a fiber sample capacity to hold water that is important for lignocellulosic fibers including thermal insulation materials made of them. Raw and foamed SE-LCB samples were characterized by WRV following the standard (SCAN-C 62:00) methodology [27]. Raw samples were chopped additionally to pass a 6 mm sieve to place them inside the test tubes. First, all samples were immersed in water for 24 h, then drained, placed in the dry glass tubes and centrifuged at 4500 rpm for 15 min to eliminate the excess water. Then, the tubes were oven-dried overnight at 105 °C, weighing the mass before (m_1_) and after (m_2_) the drying. After the deduction of the tube mass, the WRV is calculated in percentages from the equation:WRV = ((m_1_/m_2_) − 1) × 100.(2)

#### 2.4.6. Determination of Bulk Density

Bulk density of raw and foamed SE-LCB samples was measured using a standard (EN 15103) methodology [28]. The conditioned samples were placed in the cylindrical metal container of 5 L in a loose-fill way, and its bulk density (D_ar_, kg m^−3^) was calculated from the equation:D_ar_ = (m_2_ − m_1_)/V,(3)
where m_1_ is the mass of the empty container in kg; m_2_ is the mass of the container filled with a sample in kg; V is the volume of the container in m^3^.

#### 2.4.7. Determination of Thermal Conductivity

Thermal conductivity of raw and foamed SE-LCB samples was performed following the standard (ISO-8301) methodology [29] in a LINSEIS Heat Flow Meter 200 (Robbinsville, NJ, USA). The samples were placed in an open box with inner measurements of 200 × 200 × 50 mm, and the thermal conductivity was determined between the temperature range of 0 °C on the bottom plate and 20 °C on the top plate. The thickness of all samples was constant (50 mm) while the density varied from bulk (loose-fill) to set values of 30, 45, 60, 75 and 90 kg m^−3^. The density range of the investigated LCB samples was selected based on the typical cellulose-based loose-fill thermal insulation material [30] and the maximal bulk density that was able to be achieved in a loose-fill way.

### 2.5. Statistical Analysis

The factors of the influence on the mean values of the tested properties mentioned in Section 2.4 were analyzed by Excel software using the one-way ANOVA tool at the significance level α = 0.05 [31].

## 3. Results

### 3.1. Chemical Composition and FTIR Analysis of Used LCB

Results of detected chemical components of raw and pretreated LCB samples (% of oven dry mass) are shown in Figure 2. The average ash content is very similar for WS-raw and Reed-raw samples (3.9% and 4.3%, respectively), which significantly differ from the CS-raw sample (8.5%). The ash content in all used LCB significantly decreased after the SE pretreatment. The cellulose content in the form of glucan significantly varies in the used raw LCB from 30.3% (CS-raw) to 39.9% (WS-raw) that significantly increases in all samples after the SE pretreatment.

The hemicelluloses content (sum of xylan, galactan, arabinan, mannan and acetyl groups) of used raw LCB varies in a lower extent from 23.9% (CS-raw) to 29.9% (Reed-raw). Typically, the hemicellulose content in LCB decreases after the SE pretreatment [14] as was detected also in our study for WS and reed samples; however, the hemicellulose content (namely xylan) of corn stalk was slightly increased, indicating a specific chemical structure of the species that could be attributed to the relatively mild SE conditions. The detected lignin content of raw LCB samples varies also in a lower extent from 19.4% (WS-raw) to 24.2% (Reed-raw). The lignin content in LCB significantly increased after the SE treatment for WS and reed samples, while it slightly decreased in corn LCB, again, indicating to the different corn structure.

For the comparison, the detected chemical components of raw and SE-reed were reported by Lizasoain et al. [32], demonstrating cellulose, hemicellulose and lignin contents of 42.8%, 26.6% and 14.6%, respectively, for raw samples and 44.5%, 5.6% and 12.9% for SE samples treated at the same severity factor of logR0 = 3.53. These differences of raw samples may refer to the soil and growing conditions as well as to some different details conducting the determination of the chemical components.

The contents of cellulose, hemicelluloses and lignin of raw corn stalk and wheat straw were summarized by other authors [4] in the range of 35.0–39.6%, 16.8–35.0% and 7.0–18.4% and of 32.9–50.0%, 15.0–35.5% and 5.2–20.0% (including ash of 0.8–5.9%), respectively, which fit quite well our obtained values.

The FTIR absorption bands of raw and SE-LCB samples were assigned in accordance with the references [33,34], and the results are summarized in Figure 3. As was mentioned above, plant biomass is composed mainly of polysaccharides (cellulose and hemicelluloses) and lignin and a relatively small amount of various minor compounds. All spectra have the mayor absorption maxima around 1050 cm^−1^ typical for various C-O bond deformations in secondary and primary alcohols mainly originated from polysaccharides.

Spectra of biomass before and after SE treatment showed similarity of absorption bands, but their intensities differ. This means that SE treatment performed at 230 °C for 30 s did not raise significant changes in biomass composition. The broad absorption band of the OH group stretching at 3400 cm^−1^ and the C–H symmetric and asymmetric C–H stretching around 2920 cm^−1^ is similar for all samples. The difference between FTIR spectra was found in the carbonyl group region and fingerprint region. The absorption maximum of the carbonyl group (C=O) at 1734 cm^−1^ and C–O bond stretch of 1250 cm^−1^, typical for acetate esters, decreased after SE treatment that reveals a particular cleavage of ester linkage and deacetylation of hemicelluloses by SE treatment. The biomass after SE treatment contains a smaller amount of acetate groups which was confirmed also by chemical analysis.

The lignin is an aromatic biopolymer, and it showed many typical absorption bands of aromatic rings at 1600 cm^−1^, 1515 cm^−1^ and 1420 cm^−1^; the absorption band of aliphatic C–H was mainly of methoxyl group at 1460 cm^−1^. The presence of absorption bands at ~1320 cm^−1^ and 834 cm^−1^ confirm the presence of syringyl and para-hydroxyphenyl groups in the lignin of biomass [34]. The lignin is a more recalcitrant macro compound than hemicelluloses; therefore, its relative content increased after SE was approved by Jakobsons et al. [35]. Comparison of absorption intensities at 1515 cm^−1^ (lignin aromatic ring) and 1050 cm^−1^ (mainly polysaccharide origin alcohols) showed that the relative content of lignin did not change significantly but in the case of SE-corn even decreased (see also Figure 2). FTIR spectra of SE wheat straw [36] and giant reed [37] detected by other authors indicated similar tendencies, however, depending on the severity factor. For example, at the severities of logR0 ≥ 4.15, the lignin concentration highly increases due to the formation of so called pseudolignin [35]. While in our case, with logR0 values between 3.23 and 3.75, only lignin depolymerization reactions occurred, resulting in its leaching into the case of corn the sample.

### 3.2. Bulk Density and Particle Size Distribution

The results of bulk density and particle size distribution of raw and SE-treated WS, reed and CS depending on processing are summarized in Figure 4. It should be noted that moisture content of detected conditioned raw samples varied about 9.8 ± 0.4% while for the treated samples it varied—7.1 ± 0.8%. The less moisture content of SE-LCB fits the less content of hemicelluloses (Figure 2) which are responsible for moisture and biological degradation of lignocellulosic material [38].

As seen from the results (Figure 4), the bulk density of all used raw LCB is significantly higher than of treated ones; this indicates that the effective pretreatment of SE is capable of disrupting the raw lignocellulosic structure into the fibrous one and of enhancing the specific density of bulk material capable of filling a higher area. However, it should be noted that in the framework of this study, the lower bulk density of SE-LCB was positively influenced also by the foaming process through the system of stainless-steel wires (Figure 1).

The lowest bulk density for all SE-LCB was achieved after 30 s of the treatment indicating the optimal SE condition; however, the significant difference between the density values was detected only for the treated WS and reed samples. The lowest bulk density between the SE-LCB was detected for wheat straw sample (17 kg m^−3^) that dropped even by 65% compared to the raw bulk density (49 kg m^−3^). For reed samples, the highest drop in bulk density was accounted by 61%, but for corn stalks it was only 25%. This phenomenon could be related to the structural differences of each LCB including chemical composition that was only slightly changed for WS sample under the SE conditions compared with reed and CS samples, respectively (Figure 2). For example, Schnabel et al. [17] reported that the bulk density of corn straw steam exploded under the conditions of 200 °C for 20 min (logR0 = 4.25) and increased from 49 kg m^−3^ (raw) to 83 kg m^−3^, pointing out the significance of the treatment conditions.

The detected particle size distribution differs significantly between the samples, indicating structural differences within the used LCB (Figure 4). In general, the SE treatment resulted in an increase of the particle fractions lower than 2 mm and higher than 10 mm. The fraction lower than 2 mm increases from 20% (raw sample) to 38% (at SE-230/40) for WS samples, from 21% to 46% (at SE-230/15 and SE-230/50) for reed samples and from 12% to 49% (at SE-230/15 and SE-230/30) for CS samples. Despite the fraction <2 mm increasing, it positively influenced the bulk density, allowing it to achieve low values as mentioned before. The particle fraction of >10 mm is the main and higher one for all samples except for Reed-SE-230/50 where it is decreased to 18%. The decrease of the highest fraction indicates the significant impact of the SE treatment expressed by the severity of logR0 = 3.75 that was approved by the microscopy observations described in Section 3.4. The decrease of particle size during the SE treatment was approved in our previous study [39] as well as by other authors [17,36] pointing out the dependence on lignocellulosic origin, its preparation and SE conditions. In turn, the highest percentage of the particle fraction >10 mm of the treated samples corresponds to the lowest bulk density of the samples; this is very important not only as for thermal insulation materials, but also as for construction materials [4]. This also indicates the effective treatment of LCB that is expressed by disintegration of rough raw material to fibers with high curl index that is observed in Figure 5.

### 3.3. Analysis of Light Microscopy

The results of the microscopical view of raw and SE-LCB are shown in Figure 5. In general, the magnified view reveals the treatment effect along with increasing SE severity, in our case, with increasing time from 15 to 50 s at 230 °C. Comparing the raw samples with treated ones, the clear trend is observed for all lignocellulosic species that is expressed by a deeper defibration effect. The samples treated at 230/15 (Figure 5b) demonstrate the disrupted structure of initial raw samples that is expressed by a rough defibration forming fiber bundles with very low number of single fibers. The defibration increases with increasing SE severity that is expressed by increased number of single fibers; these are observed in agglomerates particularly after 40 s and 50 s of the treatment (Figure 5d,e). There is observed the curl shape of defibrated fibers that is a typical result of a low consistency defibration process [40] which could be attributed also to the SE process. For example, WS-SE even at a similar severity (logR0 = 3.72) but for a longer time (6 min at 200 °C) resulted in a biomass which looks like a powder, with the main fraction (40%) of fiber length being 0.5–1 mm [36]. In our case, the view of SE-LCB is very suitable for thermal insulation materials as it looks like wool. This view is a successful result of coupled processing of raw LCB by a rough chopping, SE treatment and mechanical foaming.

### 3.4. Analysis of Water Retention Values

The detected WRVs of raw and treated samples vary significantly in the range of 84–195% and depend mainly on the lignocellulosic species (Figure 6). The lowest WRV (84.3%) demonstrates the sample of raw reed indicating the lowest capacity to hold water. After the reed treatment at a short SE time (15 s), the WRV increases significantly to 97% but then decreases again with the time increase achieving the raw sample level (85%) after 40 s. The increase of WRV after short treatment time or lower severity could be related to the rise in pore volume of SE fibers [15], while the decrease of WRV at higher severity could be related to the decrease of hemicelluloses and increase of lignin [36]. The highest capacity to hold water (195.6%) showed the sample of raw corn stalk which significantly decreases after the SE treatment to the minimum (144%) achieved after 50 s. The raw WS sample demonstrates a lower WRV than raw corn; however, it increases after 15 s treatment and decreases again down to a minimum of 130% after 50 s (Figure 6). A similar WRV of refined wheat straw was reported in range of 128–154% depending on the defibering degree [9]. Besides the variation of WRV of the same species depending on treatment conditions, it was reported that the WRV of a same pulp fiber sample can vary between 90% and 185% depending on the measurement details like centrifugal time, speed and filter size [41].

Analyzing the WRV results in the framework of this study, the lowest values of SE-reed samples could be attributed to the lowest content of hydrophilic hemicelluloses and to the highest content of hydrophobic lignin, because these components are higher and lower, respectively, for wheat straw and corn samples (Figure 2). Finally, for the comparison it could be said that the WRV of industrial fibers used for thermal insulation purposes, such as thermo-mechanical softwood fiber and ecowool produced from recycled paper, was detected as high as 651 ± 57% and 714 ± 73%, respectively. Therefore, the WRV results of our investigated thermal insulation materials are quite good since a lower water hold value is more favorable than higher one.

### 3.5. Analysis of Thermal Conductivity vs. Bulk Density

The values of thermal conductivity of raw WS, reed and CS samples at the bulk density level (Figure 4) were detected in range of 0.0431 ± 0.0014 W m^−1^ K^−1^, 0.0462 ± 0.0011 W m^−1^ K^−1^ and 0.0432 ± 0.0023 W m^−1^ K^−1^, respectively, depending on the chopping fraction (6 mm, 10 mm, 20 mm and 30 mm). The lowest values demonstrate the samples chopped at the fraction of 10 mm; however, the fractions have the highest bulk density of 77 kg m^−3^, 114 kg m^−3^ and 68 kg m^−3^, respectively, for raw WS, reed and CS samples.

The results of the thermal conductivity of WS samples depending on treatment time and density (D) are summarized in Figure 7. As it is seen, the relationship between thermal conductivity and sample density describes good (60–91.6%) polynomial regression of the second order. The thermal conductivity depending on the density of WS-SE-230/15 samples vary in a range of 0.0429–0.0468 W m^−1^ K^−1^ (average values) with the lowest value achieved at the density of 60 kg m^−3^ which significantly differs only from the sample with the density of 30 kg m^−3^; 91.57% of the relationship is described by the polynomial regression in Equation (4).

The thermal conductivity depending on the density of WS-SE-230/30 samples varies in range of 0.0430–0.0451 W m^−1^ K^−1^ with the lowest value achieved at the density of 90 kg m^−3^ which significantly differs only from the sample with a density of 30 kg m^−3^; 59.96% of the relationship is described by the polynomial regression in Equation (5).

The thermal conductivity depending on the density of WS-SE-230/40 samples vary in the range of 0.0420–0.0456 W m^−1^ K^−1^ with the lowest value achieved at the density of 60 kg m^−3^ which significantly differs only from the sample with a density of 90 kg m^−3^; 89.2% of the relationship is described by the polynomial regression in Equation (6).
λ_WS-SE-230/15_ = 0.000002(D^2^) − 0.0003(D) + 0.0539.(4)
λ_WS-SE-230/30_ = 0.0000007(D^2^) − 0.0001(D) + 0.0475.(5)
λ_WS-SE-230/40_ = 0.000003(D^2^) − 0.0003(D) + 0.0512.(6)
λ_WS-SE-230/50_ = 0.000002(D^2^) − 0.0002(D) + 0.0471.(7)

The thermal conductivity depending on the density of WS-SE-230/50 samples varies in the range of 0.0412–0.0443 W m^−1^ K^−1^ with the lowest value achieved at the density of 45 kg m^−3^ which significantly differs only from the sample with a density of 90 kg m^−3^; 74.9% of the relationship is described by the polynomial regression in Equation (7).

The results of the thermal conductivity of reed samples depending on SE mode and density are summarized in Figure 8. The thermal conductivity depending on the density of Reed-SE-230/15 samples varies in the range of 0.0465–0.0536 W m^−1^ K^−1^ (average values) with the lowest value achieved at a density of 60 kg m^−3^ which significantly differs only from the sample with a density of 30 kg m^−3^; 83.4% of the relationship is described by the polynomial regression in Equation (8).

The thermal conductivity depending on the density of Reed-SE-230/30 samples varies in the range of 0.0427–0.0452 W m^−1^ K^−1^ with the lowest value achieved at the density of 60 kg m^−3^ which significantly does not differ from other samples values; therefore, only 52.9% of the relationship is described by the polynomial regression in Equation (9).

The thermal conductivity depending on the density of Reed-SE-230/40 samples varies in the range of 0.0427–0.0444 W m^−1^ K^−1^ with the lowest value achieved at the density of 60 kg m^−3^ which significantly does not differ from other samples like in the case of Reed-SE-230/30 samples; 60.75% of the relationship is described by the polynomial regression in Equation (10).

The thermal conductivity depending on the density of Reed-SE-230/50 samples varies in the range of 0.0406–0.0469 W m^−1^ K^−1^ with the lowest value achieved at the density of 30 kg m^−3^ and the highest value achieved at the density of 90 kg m^−3^ which significantly differs from all other samples; 91.14% of the relationship is described by the polynomial regression in Equation (11).
λ_Reed-SE-230/15_ = 0.000004(D^2^) − 0.0005(D) + 0.0671.(8)
λ_Reed-SE-230/30_ = 0.000002(D^2^) − 0.0002(D) + 0.0504.(9)
λ_Reed-SE-230/40_ = 0.0000006(D^2^) − 0.00005(D) + 0.0438.(10)
λ_Reed-SE-230/50_ = 0.000003(D^2^) − 0.0002(D) + 0.0457.(11)

The results of the thermal conductivity of corn stalk samples depending on SE mode and density are summarized in Figure 9. The thermal conductivity depending on the density of Corn-SE-230/15 samples varies in the range of 0.0437–0.0489 W m^−1^ K^−1^ with the lowest value achieved at the density of 60 kg m^−3^ which significantly differs only from the sample with the density of 30 kg m^−3^; 88.3% of the relationship is described by the polynomial regression in Equation (12).

The thermal conductivity depending on the density of Corn-SE-230/30 samples varies in the range of 0.0445–0.0498 W m^−1^ K^−1^ with the lowest values achieved at a density of 60–75 kg m^−3^ which significantly differ only from the sample with a density of 30 kg m^−3^; 82.9% of the relationship is described by the polynomial regression in Equation (13).

The thermal conductivity depending on the density of Corn-SE-230/40 samples varies in the range of 0.0448–0.0522 W m^−1^ K^−1^ with the lowest value achieved at a density of 60 kg m^−3^ and the highest value achieved at a density of 30 kg m^−3^ which significantly differs from all other samples; 88.3% of the relationship is described by the polynomial regression in Equation (14).

The thermal conductivity depending on the density of CS-SE-230/50 samples varies in the range of 0.0434–0.0488 W m^−1^ K^−1^ with the lowest values achieved at the density of 60 kg m^−3^ (however very close values in range of the density from 45 to 75 kg m^−3^) which significantly differs from the samples with the density of 30 and 90 kg m^−3^, accordingly. A total of 94.5% of the relationship is described by the polynomial regression in Equation (15).
λ_CS-SE-230/15_ = 0.000003(D^2^) − 0.0005(D) + 0.0601,(12)
λ_CS-SE-230/30_ = 0.000003(D^2^) − 0.0005(D) + 0.0607,(13)
λ_CS-SE-230/40_ = 0.000004(D^2^) − 0.0006(D) + 0.0663,(14)
λ_CS-SE-230/50_ = 0.000005(D^2^) − 0.0006(D) + 0.0633,(15)

## 4. Discussion

As was shown above, the used LCB of wheat straw, reed and corn stalk was composed mainly from cellulose for which content increased after the SE treatment in all cases. Since cellulose is the best insulator among other chemical components, this indicates the sustainable use of these available LCB in the thermal insulation applications. Moreover, the study showed an effective SE treatment during which wool-like fibrous LCB material was obtained with significantly decreased bulk density and improved thermal conductivity and WRV.

As it is shown in Section 3.5, the thermal conductivity of the investigated materials changes significantly depending on the density, and the relationship was described best by polynomial regression. The polynomial relationship between thermal conductivity and density (40–100 kg m^−3^) of differently processed flax, hemp and peat fibers was found also by other authors [8]. They declared that the lowest thermal conductivity was achieved within the density range of 70–100 kg m^−3^, depending on the used processing method and raw material. In the framework of this study, the obtained results of thermal conductivity at different density levels will serve as a prediction tool suggesting the selection of the optimal density at filling the construction cavities by the investigated loose-fill lignocellulosic material.

In general, in the framework of the study, the thermal conductivity was decreased after the defibration by SE, however, depending on the chopping fraction and SE conditions. In nine cases from twelve of our SE samples, the lowest thermal conductivity values were achieved at the density of 60 kg m^−3^. Moreover, the lowest difference in detected thermal conductivity values (0.0407–0.0465 W K^−1^ m^−1^) between all SE-LCB is also observed namely at the density level of 60 kg m^−3^. From another point of view, only the corn sample (CS-SE-230/50) from all SE-LCB achieved the lowest average thermal conductivity at the density of 60 kg m^−3^, while for wheat straw (WS-SE-230/50) and reed (Reed-SE-230/50) samples it was achieved at the densities of 45 kg m^−3^ and 30 kg m^−3^, respectively. As it is seen, the treatment at SE-230/50 resulted in the best thermal conductivity of all used LCB, however at different densities. It should be also mentioned that the difference of thermal conductivity between the samples treated at different SE conditions and measured at these densities is significant only between the reed samples treated at 50 s and 15–30 s. So, only in the case of reed sample the SE treatment is significant at 230/50, while for wheat straw and corn stover the time of treatment could be even shorter (e.g., 30 s).

The favorable results of the study encourage research to be continued providing the extended testing of the investigated LCB-based thermal insulation materials for other essential properties like settlement, water vapor diffusion, reaction to fire, mold fungi resistance and volatile organic compounds; these all are intended to be conducted in the near future.

## 5. Conclusions

Based on the study results the following conclusions are derived:Lignocellulosic biomass of wheat straw, water reed and corn stalk treated by steam explosion under the conditions of 230 °C for 30 s and mechanically foamed by rotating wires system achieves significantly decreased bulk density (17–28 kg m^−3^) and contains mainly of cellulose (40–46%) that makes it suitable for thermal insulation applications.The lowest bulk density was attributed to the highly curled particle dimensions comprising the main fraction over 10 mm between 30% (corn) and 69% (reed).Water retention values of the treated biomass vary significantly (85–172%), with the lowest ones demonstrating reed samples, and have a tendency to decrease for all species with increasing SE severity that is attributed to the loss of hemicellulose.The relationship between detected thermal conductivity and bulk density of the samples in the range of 40–100 kg m^−3^ was described best (R^2^ 53–94%) by the polynomial regression equations which approved the optimal thermal conductivity values (0.0409–0.0439 W K^−1^ m^−1^) achieved at the density between 45 kg m^−3^ and 60 kg m^−3^ for the samples treated for 50 s.This study suggests that the density level of investigated loose-fill lignocellulosic species not be higher than 60 kg m^−3^ in cases of application in the building cavities in a blow-in way.

## Figures and Tables

**Figure 1 materials-16-03654-f001:**
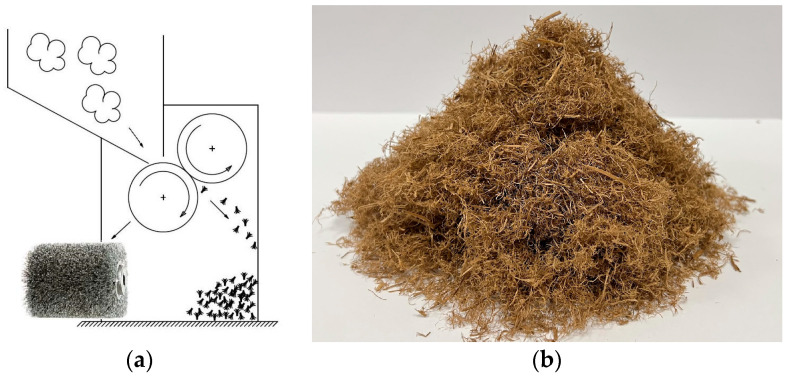
(**a**) Fiber foaming system for SE-LCB; (**b**) foamed SE-LCB sample (WS-SE-230/30) before testing.

**Figure 2 materials-16-03654-f002:**
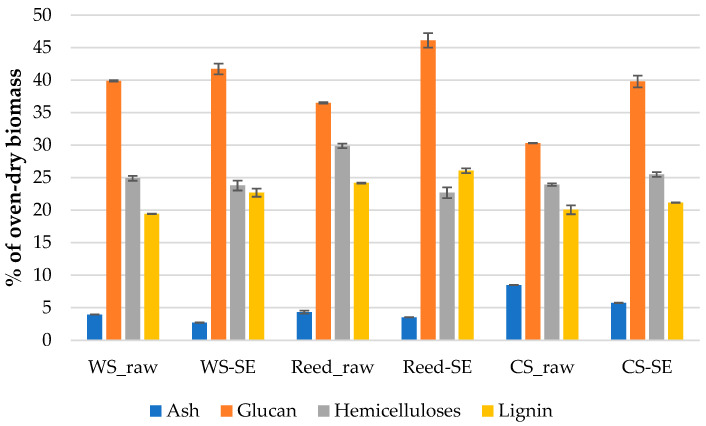
Chemical composition of raw and treated (SE-230/30) LCB.

**Figure 3 materials-16-03654-f003:**
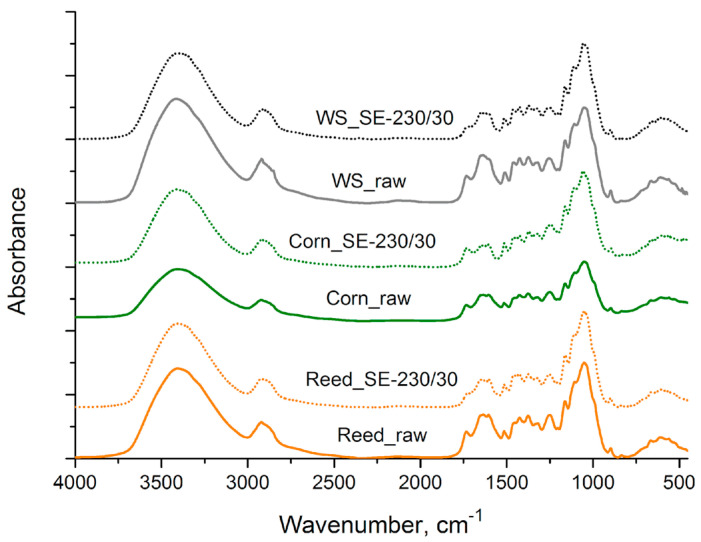
FTIR spectra of raw and SE-LCB.

**Figure 4 materials-16-03654-f004:**
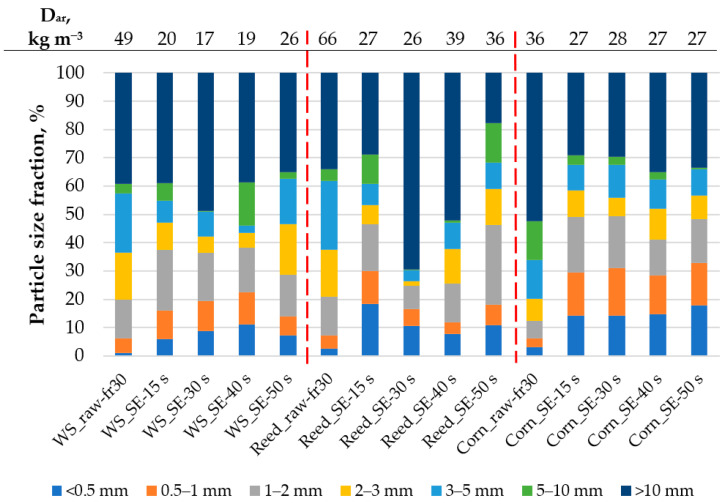
Bulk density (D_ar_) and distribution of LCB particle size depending on the processing. The dashed line divides different LCB.

**Figure 5 materials-16-03654-f005:**
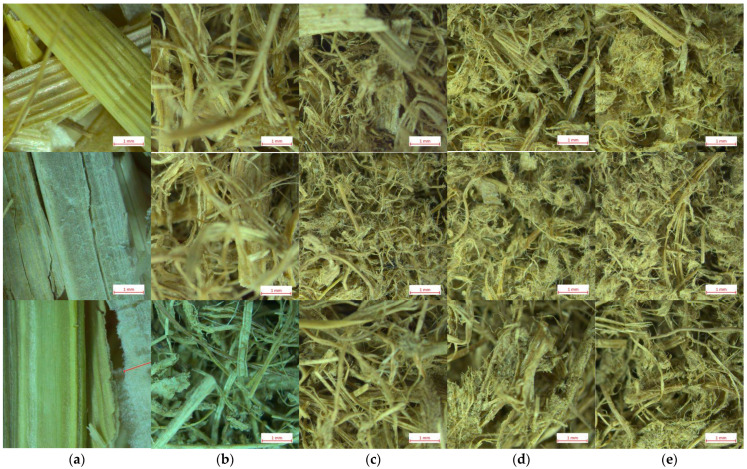
Microscopical surface view (0.63×, scale bars 1 mm) of wheat straw (**above**), reed (**middle**) and corn stalk samples (**bottom**): (**a**) Raw and steam-exploded at 230 °C after (**b**) 15 s, (**c**) 30 s, (**d**) 40 s and (**e**) 50 s.

**Figure 6 materials-16-03654-f006:**
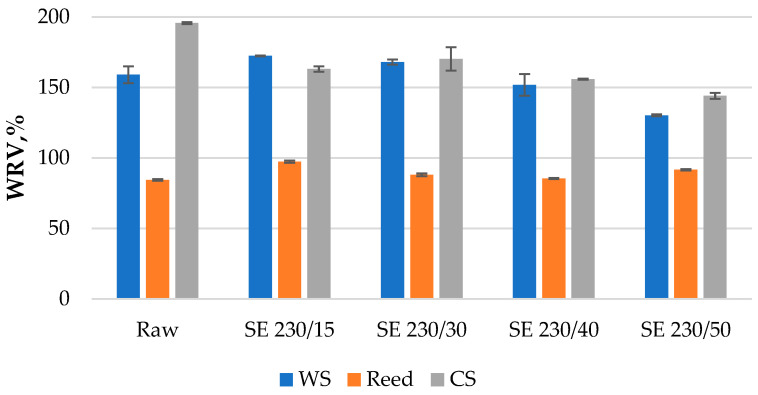
Water retention values of raw and foamed SE-LCB.

**Figure 7 materials-16-03654-f007:**
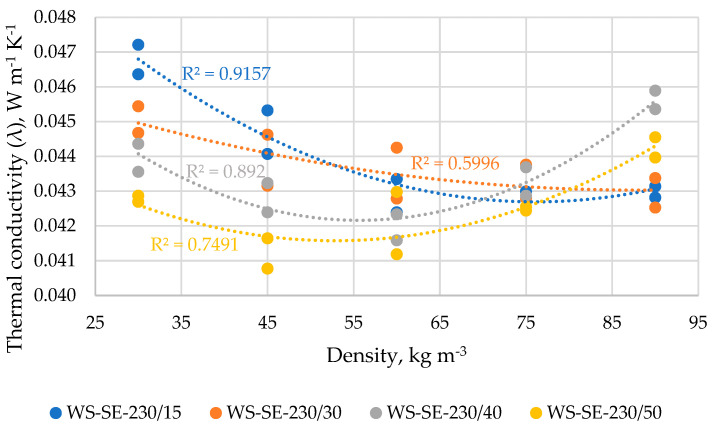
Thermal conductivity vs. density of wheat straw samples depending on SE conditions expressed in polynomial curves of the second order.

**Figure 8 materials-16-03654-f008:**
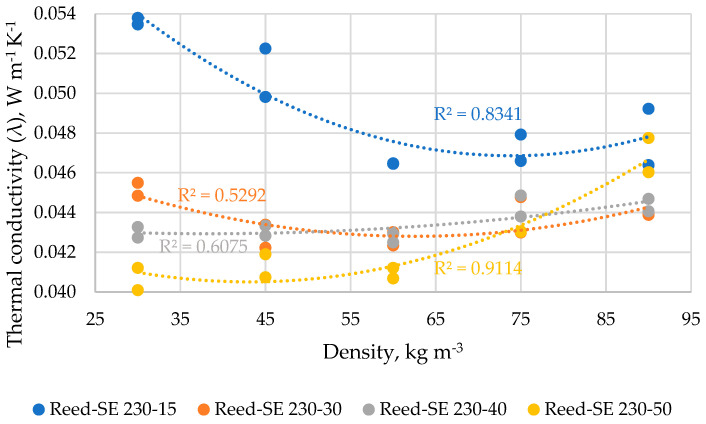
Thermal conductivity vs. density of reed samples depending on SE conditions expressed in polynomial curves of the second order.

**Figure 9 materials-16-03654-f009:**
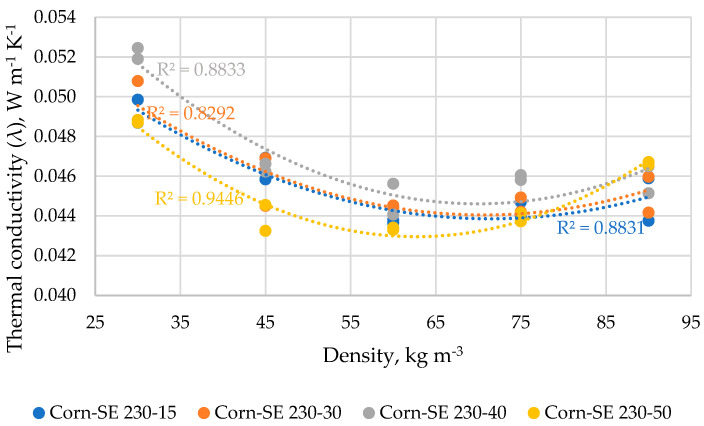
Thermal conductivity vs. density of corn stalk samples depending on SE conditions expressed in polynomial curves of the second order.

**Table 1 materials-16-03654-t001:** Experimental design of steam explosion for each raw LCB.

Sample	Time, s	logR0
LCB ^1^-SE-230 ^2^/15	15	3.23
LCB-SE-230/30	30	3.53
LCB-SE-230/40	40	3.65
LCB-SE-230/50	50	3.75

^1^ LCB—appropriate lignocellulosic biomass, e.g., WS, reed or CS; ^2^ 230—temperature used at SE treatment.

## Data Availability

The raw data presented in this study are available on request from the corresponding author.

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
