# Peer review of "Optimization of Thermal Conductivity vs. Bulk Density of Steam-Exploded Loose-Fill Annual Lignocellulosics"

_materials, 2023, doi:10.3390/ma16103654_

Round 1
Reviewer 1 Report
The objective of this manuscript was to explore new thermal insulation materials from locally available residues of annual plants like wheat straw, reeds and corn stalks after obtaining fibres by mechanical crushing and defibration by steam explosion process.
The title is not adequate.
The novelty of the paper is not high, since there are studies in the literature to explore different types lignocelullosic biomass for thermal insulation materials using different processes to obtain the fibres. However, this is a complete study that compares different annual plants residues in a very systematic way.
The sentence “The pretreatment of raw materials was performed by mechanical crushing and defibration by steam explosion process” is not in my point of view correct. In fact the so called pretreatment is the process to obtain the fibers from the biomass. For example in case of the manufacture of paper, the thermomechanical pulping is not called a pretreament.
The state-of-the-art section gives an overview of the works published in the literature about the theme.
In materials and methods, the methods are adequate and in general the relevant information is provided, except in case of steam explosion device, for which information should be provided and the determination of moisture content should be explained.
The discussion of results is very complete and the general conclusions are well supported by the results.
Some revisions are also indicated in the manuscript. So, I think that the paper should be accepted with minor revisions.

Author Response
Thanks for your valuable review and suggestions to improve our manuscript!
The objective of this manuscript was to explore new thermal insulation materials from locally available residues of annual plants like wheat straw, reeds and corn stalks after obtaining fibres by mechanical crushing and defibration by steam explosion process. Response: Thank you!
The title is not adequate. Response: The manuscript actually is focused on investigation of thermal conductivity of loose-fill LCB depending on varying density; devoting around 50% of the manuscript. Taking into account that this is the third manuscript of authors devoted to the investigation of selected LCB on thermal insulation, the title is adequate enough in our opinion.
The novelty of the paper is not high, since there are studies in the literature to explore different types lignocelullosic biomass for thermal insulation materials using different processes to obtain the fibres. However, this is a complete study that compares different annual plants residues in a very systematic way. Response: Thank you!
The sentence “The pretreatment of raw materials was performed by mechanical crushing and defibration by steam explosion process” is not in my point of view correct. In fact the so called pretreatment is the process to obtain the fibers from the biomass. For example in case of the manufacture of paper, the thermomechanical pulping is not called a pretreament. Response: The term “pretreatment” was changed to “treatment” in the sentence and throughout the manuscript.
The state-of-the-art section gives an overview of the works published in the literature about the theme. Response: Thank you!
In materials and methods, the methods are adequate and in general the relevant information is provided, except in case of steam explosion device, for which information should be provided and the determination of moisture content should be explained. Response: the details of SE device and determination of moisture content were added.
The discussion of results is very complete and the general conclusions are well supported by the results. Response: Thank you!
Some revisions are also indicated in the manuscript. So, I think that the paper should be accepted with minor revisions. Response: Thank you, the revisions indicated in the manuscript were amended.
Reviewer 2 Report
Dear Authors,
I found your manuscript as well prepared and of high potential to readers due to the topic, of the valorization of lignocellulosic raw materials.
However, below please find just a few remarks that should improve your manuscript before publication:
- in the Introduction paragraph, I suggest including more recent publications concerning the bulk density of lignocellulosic materials - doi: 10.3390/ma14247772
- fig. 2, the plot legend: I suggest unifying the spelling of the word "hemicelluloses" (as is in the text) but here in the plot is "hemicelulozes"
- the chapter "4. Discussion" should be divided into "4. Discussion" and "5. Conclusion", where some of the paragraphs from the current "4. Discussion" should be moved and developed
- 2nd paragraph in "4. Discussion", the word "the density in the rage of..." should be changed to "the density in the range of..."
Best regards!
Author Response
Thanks for your valuable review and suggestions to improve our manuscript!
- in the Introduction paragraph, I suggest including more recent publications concerning the bulk density of lignocellulosic materials - doi: 10.3390/ma14247772. Response: thanks for the suggestion, however, in our opinion just the comparison of bulk density of various lignocellulosics would not contribute to the state of the art of the manuscript since it deals with thermal insulation materials, particularly, with thermal conductivity; in turn, the suggested paper include information about bulk density of lignocellulosics used for particleboards that it is not related to our topic. However, this section was supplemented by some citations highlighting the importance of relationship between material density and thermal conductivity.
- fig. 2, the plot legend: I suggest unifying the spelling of the word "hemicelluloses" (as is in the text) but here in the plot is "hemicelulozes". Response: Thank you, the word was corrected.
- the chapter "4. Discussion" should be divided into "4. Discussion" and "5. Conclusion", where some of the paragraphs from the current "4. Discussion" should be moved and developed. Response: we have chosen to exclude the Conclusion section since it includes the Discussion section. Besides, such a structure supports the journal guidelines (https://www.mdpi.com/journal/materials/instructions): “Conclusions: This section is not mandatory but can be added to the manuscript if the discussion is unusually long or complex.” However, we modified the title of the section as “Discussion and Conclusion”.
- 2nd paragraph in "4. Discussion", the word "the density in the rage of..." should be changed to "the density in the range of...". Response: Thank you, the word was corrected.
Reviewer 3 Report
The study deals with the investigation of steam-exploded loose-fill insulation. Different plant materials, process parameters and material properties were investigated.
The topic is interesting and important for different stakeholders. The manuscript is well structured. However, not all-important properties or challenges in the application have been addressed. Therefore, this manuscript cannot be recommended for publication in the journal Materials without modifications.
further remarks
Normally, effects on the wax layer should also be evident in the FT-IR measurements. Did you notice any changes here?
Please note that you have only analysed a small bulk density range for your investigations regarding thermal conductivity. A larger range could possibly also show a different model, please also make it clear in the manuscript that your model does not have a generally valid statement in the range above 90 kg/m³ or prove it.
Furthermore, there are no conclusions in this manuscript. Please be sure to add a conclusion.
For the application, not only the thermal conductivity and the bulk density are interesting/important, but also the settlement behaviour of the loose-fill insulation. Therefore, an estimation of the settlement behaviour would be important here. Investigations on a vibrating plate would be possible here. Do you have such a possibility for these tests?
The results of the different energy consumption of the individual treatments would also be interesting. Could you add some results here?
Author Response
Thanks for your valuable review and suggestions to improve our manuscript!
Normally, effects on the wax layer should also be evident in the FT-IR measurements. Did you notice any changes here? Response: We measured FTIR spectra in KBr pellets and in this case we did not observe any special evidence of changes in waxes layer. Probably because concentration of waxes could be quite low in biomass under study and FTIR absorption maxima of aliphatic part or ester linkage can not be separated of other lipophilic compounds.
Please note that you have only analysed a small bulk density range for your investigations regarding thermal conductivity. A larger range could possibly also show a different model, please also make it clear in the manuscript that your model does not have a generally valid statement in the range above 90 kg/m³ or prove it. Response: The density range we selected up to 90 kg/m3 because it is not possible to get higher for loose-fill materials nor manually, nor using blow-in equipment. The exception is raw lignocellulosics with a natural higher density like in the case of reeds, however, it is also dependent on the shredding fraction. This clarification was added in the Section 2.4.7. Since the investigation was performed within the defined density range it’s clear that the obtained models are valid only within the defined range.
Furthermore, there are no conclusions in this manuscript. Please be sure to add a conclusion. Response: we have chosen to exclude the Conclusion chapter since it includes the Discussion section. Besides, such a structure supports the journal guidelines (https://www.mdpi.com/journal/materials/instructions): “Conclusions: This section is not mandatory but can be added to the manuscript if the discussion is unusually long or complex.” However, we modified the title of the section as “Discussion and Conclusion”.
For the application, not only the thermal conductivity and the bulk density are interesting/important, but also the settlement behaviour of the loose-fill insulation. Therefore, an estimation of the settlement behaviour would be important here. Investigations on a vibrating plate would be possible here. Do you have such a possibility for these tests? Response: Thank you for the suggestion about settlement behavior of our investigated materials. As you know, it is impossible and not necessary to perform and analyze all important properties of a material within one manuscript. However, as we already noted in the end of Discussion section some future works are proposed including settlement test.
The results of the different energy consumption of the individual treatments would also be interesting. Could you add some results here? Response: Thank you for the suggestion. Sure, the energy consumption is very relevant, however, at this moment we have no the results. From another point of view, to discuss about the energy consumption would be possible only in the case of industrial scale study. Since we have performed the study based on laboratory equipment it would be just a speculation to discuss about it.
Reviewer 4 Report
The manuscript: “Optimization of Thermal Conductivity vs Bulk Density of Steam-Exploded Loose-Fill Annual Lignocellulosics” investigates new thermal insulation materials from locally available residues of annual plants like wheat straw, reeds and corn stalks. Optimization of thermal conductivity of the obtained loose-fill thermal insulation materials was investigated at different bulk density levels. The obtained results suggest the adjustment of density to achieve an optimal thermal conductivity of LCB-based thermal insulation materials, the study is interesting, however it is necessary to make the following corrections:
1. In the section on thermal conductivity as a function of densities, it is necessary to reconsider both the presentation of results and the discussion. It is very tedious to read, practically the same for each case, only the trend of the curves changes. Propose something more attractive and that the results can be understood more easily.
2. in table 1, a column for the same value (temperature) is not necessary, indicate it in the table description. In Figure 4, maybe use 1-2 mm instead 1…2 mm. In Figure 5, the legend of the magnitude on the scale is too small.
3. How the foaming process occurs?, please describe it.
4. What does it means; Chemical components like sugars (C-5 and C-6), C-5 and C-6?
5. How the chemical composition was calculated in Fig 2.
6. please include a general conclusion in the manuscript.
Minor editing of English language required
Author Response
Thanks for your valuable review and suggestions to improve our manuscript!
- In the section on thermal conductivity as a function of densities, it is necessary to reconsider both the presentation of results and the discussion. It is very tedious to read, practically the same for each case, only the trend of the curves changes. Propose something more attractive and that the results can be understood more easily. Response: Thanks for your suggestion, however, as we know the scientific language is tedious almost ever. The main goal of the section 3.5 was to prove that the thermal conductivity is essential on the sample density and based on it to suggest an optimal density range for investigated thermal insulation materials for application in a building cavity. In our opinion we presented the data quite understandable, at a minimal manner, providing an explanation supported by graphs and formulas. Even if to reconsider the presentation style the results will not change. Taking into account that the presented style of results in the chapter was appreciated without objections by other reviewers, we keep rights to stay the section at the original submission version.
- in table 1, a column for the same value (temperature) is not necessary, indicate it in the table description. In Figure 4, maybe use 1-2 mm instead 1…2 mm. In Figure 5, the legend of the magnitude on the scale is too small. Response: Thank you, all the recommendations were corrected.
- How the foaming process occurs?, please describe it. Response: The process is described in section 2.3, where we added small clarification. Actually, the mechanical processing of wet squeezed fiber mass through the wired cylinders we call foaming, because the fiber mass gets fluffy and reminds a foam.
- What does it means; Chemical components like sugars (C-5 and C-6), C-5 and C-6? Response: We agree. Instead of "C-5 and C-6 sugars" it could be better to use pentoses (sugars with 5 carbon atoms), namely, xylose and arabinose, and hexsoses (sugars with 6 carbon atoms), namely, glucose, galactose and mannose. This part was modified in section 2.4.1.
- How the chemical composition was calculated in Fig 2. Response: The details are outlined in section 2.4.1.
- please include a general conclusion in the manuscript. Response: we have chosen to exclude the Conclusion section since it includes the Discussion section. Besides, such a structure supports the journal guidelines (https://www.mdpi.com/journal/materials/instructions): “Conclusions:This section is not mandatory but can be added to the manuscript if the discussion is unusually long or complex.” However, we modified the title of the section as “Discussion and Conclusion”.
Round 2
Reviewer 3 Report
The manuscript has been revised and is now more clearly presented. However, further changes are needed.
The objectives still lack the interesting research question or the exciting objective of why the study was done.
What area of cellulose insulation are you addressing? The values of 30 to 90 kg/m³ cannot be correct. Please also mention material properties here (e.g. thermal conductivity).
Is Subchapter 2.5 - Statistical Analysis really necessary? The reader will not find any results on statistical tests in the manuscript.
Figure 4 should still be revised, here the allocation is not correct.
The conclusions should be a separate chapter. It is unusual to have a discussion here with literature results.
Author Response
Thanks for your valuable review and suggestions to improve our manuscript!
The objectives still lack the interesting research question or the exciting objective of why the study was done. Response: Yes, we agree that the paper have missed an exciting objective, thefore the Introduction part was supplemented by the statement: “The main purpose of the study was to find out an optimal bulk density of the investigated loose-fill LCB depending on the treatment conditions for further research providing other important properties like settling, water vapor diffusion, reaction to fire, mold fungi resistance and volatile emission”.
What area of cellulose insulation are you addressing? The values of 30 to 90 kg/m³ cannot be correct. Please also mention material properties here (e.g. thermal conductivity). Response: we do address loose-fill cellulose insulation anticipated for installation in many building cavities by a blow-in way which is capable to achieve maximal density of 80 kg/m3 because of the equipment capacity. Our study approves that there is not necessary to fill the cavities by the investigated cellulosic materials to a density level higher than 60 kg/m3 because an optimal thermal conductivity (0.0433 W/(Km)) is achieved at this level. Now the statement is added to the conclusion section as well.
Is Subchapter 2.5 - Statistical Analysis really necessary? The reader will not find any results on statistical tests in the manuscript. Response: we agree that the subsection is very short, however, yes, we think it should be included because it refers to the tested properties defined in the subsubsections 2.4.1, 2.4.3, 2.4.5 – 2.4.47 and discussed in the results showing whether the differences are significant or not. Furthermore, the journal supports the manuscript structure to be divided to the sections, subsections and even subsubsections for the better subject clarification.
Figure 4 should still be revised, here the allocation is not correct. Response: the figure was corrected.
The conclusions should be a separate chapter. It is unusual to have a discussion here with literature results. Response: the separate section of Conclusions was added.
Reviewer 4 Report
The manuscript has been improved
Author Response
Thanks for your valuable review and suggestions to improve our manuscript!
Comments and Suggestions for Authors: The manuscript has been improved Response: Thank you very much!